# One Graph to Track Them All: Dynamic GNNs for Single- and Multi-View Tracking

## Abstract

This work presents a unified, fully differentiable model for multi-people tracking that learns to associate detections into trajectories without relying on pre-computed tracklets. The model builds a dynamic spatiotemporal graph that aggregates spatial, contextual, and temporal information, enabling seamless information propagation across entire sequences. To improve occlusion handling, the graph can also encode scene-specific information. We also introduce a new large-scale dataset with 25 partially overlapping views, detailed scene reconstructions, and extensive occlusions. Experiments show the model achieves state-of-the-art performance on public benchmarks and the new dataset, with flexibility across diverse conditions. Both the dataset and approach will be publicly released to advance research in multi-people tracking.

## 1 Introduction

Multi-People Tracking is a well-researched topic where tracking-by-detection has become the *de facto* standard. Algorithms relying on Intersection-over-Union (IoU) matching to associate detections into trajectories (Zhang et al., 2022; Bergmann et al., 2019; Wojke et al., 2017) currently top state-of-the-art benchmarks. More recently, learning-based approaches such as transformer-based methods ( (Zeng et al., 2022; Meinhardt et al., 2022)) and graph neural networks ( (Cetintas et al., 2023)) have been proposed. While performance has steadily improved due to the availability of high-quality detectors, association algorithms remain highly specialized. For example, standard monocular association strategies are not equipped to handle multi-view data, and vice versa.

Single-view models often neglect to exploit the underlying scene structure and only focus on appearance and bounding-box information. Multi-view methods incorporate scene structure but often only at detection time to remove the need for association across views. Moreover, existing tracking benchmarks focus on short sequences with minimal **scene** occlusions. In contrast, real-world applications require handling long sequences with severe occlusions caused by people occluding each other and environmental structures, such as walls and columns, hiding them. These occlusions can cause extended disappearances and pose significant challenges to current tracking algorithms. While existing benchmarks feature lots of occlusions caused by people hiding each other, scene-induced occlusions remains underexplored.

In this work, we introduce a unified, fully differentiable model that replaces traditional association strategies. It learns to associate detections directly into trajectories in an online manner. Our approach builds a spatiotemporal graph where edges connect detections both temporally and across camera views in multi-view scenarios. By simultaneously aggregating appearance, motion, and contextual cues through message passing in the graph, our model enables seamless information flow across entire sequences. The graph structure is dynamically updated during both training and inference to handle streaming data in an online fashion. Our model assigns probabilities to the graph's edges and vertices to indicate whether they belong to the same person's trajectory. These probabilities are then used by an efficient online algorithm to extract the most likely trajectories.

Since existing datasets lack information about scene structure, we created our own. This new dataset is the largest of its kind, containing 25 partially overlapping views and more than 400 meters of tracked paths. It

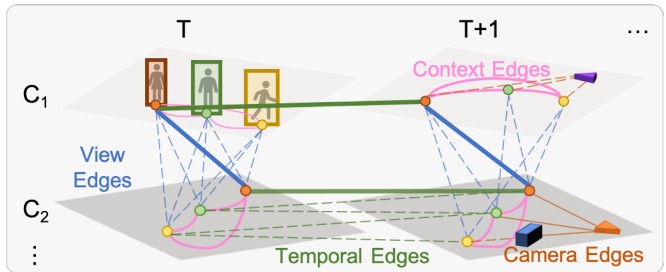

Figure 1: **Multi-View Spatio-Temporal Graph.** **View Edges** connect identical detections across views at the same timepoint. **Temporal Edges** connect detections in the same view at different timepoints, forming a fully connected graph (some connections omited for clarity). **Contextual Edges** connect detections to nearby ones in the same view at the same timepoint. **Camera vertex and edges** can be added to model scene priors and occlusions.

comprises two sequences of 20 minutes each, equivalent to 24,000 frames per camera at 10 FPS. Additionally, it provides camera calibrations and detailed scene mesh information. The dataset can be used to train and evaluate models in both single-view and multi-view scenarios, as well as to develop new methods to leverage scene structure for tracking.

Our main contributions are as follows:

1. **A unified tracking model**: A fully differentiable approach that handles both single and multi-view tracking through dynamic spatiotemporal graphs.

2. **Integration of scene priors**: Camera vertices in the graph encode scene-specific information for better occlusion handling.

3. **A new large-scale dataset**: 25 camera views featuring extensive occlusions and detailed scene reconstructions.

4. **State-of-the-art performance**: The only method achieving competitive results on both single-view and multi-view people tracking benchmarks.

Extensive experimentation shows that our approach matches the state-of-the-art in single-camera setups and far exceeds it in multi-camera ones. This is significant because most realistic surveillance scenarios involve multiple cameras, some of which have overlapping fields of views and some not. Thus, a method that does well in both cases is what is needed in practice. Our dataset showcases this in a real-world environment. Both our dataset and code are publicly available to foster further research.

## 2 Related Work

Multi-target tracking is a core computer vision task with applications to surveillance, autonomous driving, medical imaging, and retail automation (Ciaparrone et al., 2020; Yilmaz et al., 2006; Smeulders et al., 2014). Most current methods rely on tracking-by-detection (Andriluka et al., 2008), where targets detected in consecutive frames are associated to produce complete trajectories (Felzenszwalb et al., 2010; Ren et al., 2015; Girshick, 2015; Zhou et al., 2019). Despite progress, challenges remain in crowded scenes with occlusions.

**Traditional Approaches.** Traditional approaches use handcrafted association methods, including probabilistic filters like JPDA (Bar-Shalom & Fortmann, 1988) and MHT (Reid, 1979) for motion prediction, and combinatorial optimization techniques such as dynamic programming (Fleuret et al., 2008). Graph-based formulations, where nodes represent detections or fixed spatial locations, have been widely studied using k-shortest paths (Berclaz et al., 2011), network flow (BenShitrit et al., 2014; Wang et al., 2019), and multi-cuts (Tang et al., 2017; 2016). While efficient, they rely on hand-designed objective functions and affinity metrics and do not generalize well to multi-view association.

**Learning-based Methods.** Learning-based methods estimate association probabilities between detections. Early approaches used conditional random fields (Yang & Nevatia, 2012) or deep networks like CNNs (Son et al., 2017) and RNNs (Sadeghian et al., 2017; Milan et al., 2017) for pairwise affinities. Recent methods (He et al., 2021; Brasó & Leal-Taixé, 2020; Kim et al., 2022; Li et al., 2020) leverage graph neural networks (Kipf & Welling, 2016; Gilmer et al., 2017) to learn affinities directly on detection graphs, while others (Wojke et al., 2017; Ristani & Tomasi, 2018; Lv et al., 2024; Qin et al., 2024) focus on learning discriminative appearance features. However, most GNN-based trackers (Brasó & Leal-Taixé, 2020; Cetintas et al., 2023) operate offline on static graphs, leveraging future detections for association decisions. In contrast, our model is designed to make fully online decisions based only on past detections.

**Multi-view Tracking.** Multi-view tracking provides additional robustness through geometric constraints while introducing challenges in data fusion (Kamal et al., 2013). Some methods first perform per-view tracking before cross-view association (Bredereck et al., 2012; Lan et al., 2020; Nguyen et al., 2021), while others leverage geometry by estimating occupancy maps (Berclaz et al., 2011; Baqué et al., 2017) or projecting features onto a common ground plane (Hou et al., 2020; Song et al., 2021; Engilberge et al., 2023). Recent approaches (Ong et al., 2020; Cheng et al., 2023) explore associating monocular detections across both views and time, though they use a two-step process preventing joint reasoning. Our unified model simultaneously integrates inter-view, contextual and temporal cues.

**Datasets and Benchmarks.** Early datasets like PETS (Ferryman & Shahrokni, 2009) and TUD (Andriluka et al., 2008) focused on simple scenarios. Modern benchmarks like MOT Challenge (Milan et al., 2016; Dendorfer et al., 2021) provide diverse crowded sequences with standardized evaluation protocols. For multi-view scenarios, datasets like WILDTRACK (Chavdarova et al., 2018) and Campus (Fleuret et al., 2008) offer synchronized multi-camera sequences with calibration. While SCOUT has a lower density than MOT20, its challenges are of a different nature with significantly longer trajectories (on average 2.8x longer) and frequent multi-second occlusions behind structural elements. Thanks to multi-view annotation, we can track objects as they pass behind columns and walls—scenarios extremely challenging for single-view methods.

## 3 Approach

We took our inspiration from the many classical method that rely on constructing a graph to track people. While most previous graph-based methods were designed for offline tracking in either single- or multi-view settings, our approach is fully online and generalizes to both scenarios. As in (Berclaz et al., 2011; Brasó & Leal-Taixé, 2020; Zhang et al., 2008; Wang et al., 2019), we formulate the tracking problem as an edge classification task in a weighted spatio-temporal graph connecting object detections. Edge classification assigns probabilities of an edge being *active*, meaning that the two detections it links belong to the same person. This being done, we find optimal trajectories in the graph.

### 3.1 Spatio-Temporal Graph

Let us assume we are given a set of multi-view frames $F^t$ for time instants $1 \leq t \leq T$. Each one contains one or more camera images (views) $\mathbf{I}_1^t \ldots \mathbf{I}_C^t$, with $C \geq 1$. In each image $\mathbf{I}_i^t$, we run a detector that returns $D_i^t$ detections $\{d_i^{t,1}, \ldots d_i^{t,D_i^t}\}$. Our goal is to associate these detections into a set of trajectories $\mathcal{T}$ such that each trajectory $\mathcal{T}_a = \{d \mid identity(d) = a\}$ contains all the detections corresponding to person identity $a$.

Each detection $d_i^t$ is characterized by a set of features $f_d$. Those features may include bounding box coordinates $\mathbf{p}_d \in \mathbb{R}^4$, world coordinates of the detection $\mathbf{w}_d \in \mathbb{R}^3$, and its timestamp $\mathbf{t}_d \in \mathbb{N}$.

Given all the detections in all images $d_i^{t,j}$, we build a spatio-temporal graph $\mathcal{G} = (\mathcal{V}, \mathcal{E})$ depicted Fig. 1. Its vertices $\mathcal{V} = \{d_i^{t,j}\}$ are the detections. They are linked by a set of edges $\mathcal{E}$ of three different kinds:

- **Temporal Edges.** A temporal edge $e^t$ connects detections in the same camera view across time. It denotes a person moving from one location at a given time to another at a later time.

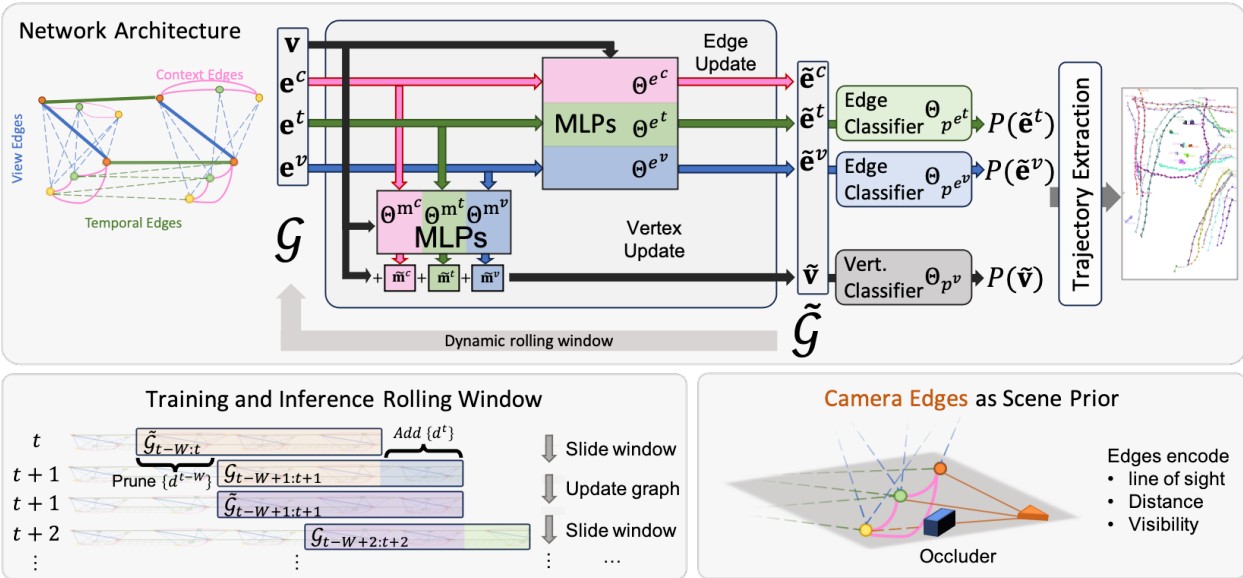

Figure 2: **Graph-based people tracking. Top.** Our UMPN network updates both vertex and edge feature vectors at each time-step and generates classification scores for the edges and vertices. These scores are used to derive the final trajectories. **Bottom left.** Dynamic graph construction using a sliding time window. **Bottom right.** Camera edges can encode environmental occlusions.

- **Multi-View Edges.** In a multi-view setup, a view edge $e^v$ connects detections in different camera views corresponding to the same time instant. It denotes the same person being seen by two different cameras.

- **Contextual Edges.** A contextual edge $e^c$ connects detections within the same camera view and time instant. It denotes a potential interaction between two people, for example, when they are near each other.

To each vertex $v \in \mathcal{V}$, we associate the detection features $\mathbf{v} \in \mathbb{R}^s$. Similarly, each edge $e \in \mathcal{E}$ is represented by a feature vector $\mathbf{e} \in \mathbb{R}^s$. Both are initialized and subsequently updated as described in Section 3.2.

**Building the Graph.** The graph starts empty and detections are added in a streaming fashion each time a new image is acquired, until it covers a window of $W$ time-steps, where $W$ is a model parameter. Then, the weighting algorithm described below classifies edges linking the same individuals. When a new frame with new detections becomes available, the temporal window is moved recursively by one time step, discarding the nodes and edges connected to the oldest frame in the graph, adding new detections, and creating new edges. The algorithms runs again on the updated graph. This is illustrated at the bottom of Fig. 2. In practice, we connect all detections within a fixed spatial and temporal distance. We provide more details in Section 5.

**Making the Graph Scene Aware.** With the growing interest in embodied AI, the need for scene understanding is becoming more and more important. In such contexts, knowledge about the scene is often available in the form of point clouds, meshes, or even semantic maps. We can use the graph structure to encode this knowledge by introducing camera vertices and edges. The camera vertices are connected to the detection vertices with camera edges. The feature vectors of the camera vertex are initialized using the camera parameters. The camera edges are used to encode the prior knowledge about the scene. They are initialized using the distance between the camera and the detection, and the angular distance between the camera line of sight and the detection location in the world coordinates.

These are optional and only used when the information required to compute the line sights—typically the camera intrinsics—-and a scene mesh can be obtained. We provide more formal definitions in the supplementary material.

## 3.2 Weighting the Edges using a GNN

Once the graph $\mathcal{G}$ has been built, our task becomes deciding which temporal and multi-view edges are active, meaning that they truly connect detections of the same person, or not. To this end, we train our Unified Message Passing Network (UMPN) to assign probabilities to these edges. The contextual edges are *not* assigned probabilities because, by construction, they connect different people. However, they convey contextual information and improve propagation of temporal and inter-view information. We also assign probabilities to the vertices to filter out false positive detections.

Similarly to previous work ((Brasó & Leal-Taixé, 2020)), our UMPN model (illustrated in Fig. 2) performs message passing among vertices. At each iteration, both the vertex and edge feature vectors are updated as described below. For simplicity, we will hereafter refer to these feature vectors as *representations*.

**Initialization.** When initially added to the graph, the edge representations are set to zero, while the vertex representations are derived from the attributes of their corresponding detections $d$, as previously described.

The attributes are normalized before being encoded using two-layer MLPs parametrized by $\Theta_{enc}$. All the encoded attributes are then concatenated to form the final representation. We write

$$\mathbf{e}^0 = \mathbf{0} \, , \tag{1}$$
$$\mathbf{v}^0 = [\mathrm{MLP}_p\left(\mathbf{p}_d\right) | \mathrm{MLP}_w\left(\mathbf{w}_d\right) | \mathrm{MLP}_t\left(\mathbf{t}_d\right)]$$

where $\mathbf{p}_d$ denotes the detection pixel coordinates, $\mathbf{w}_d$ represents its world coordinates, and $\mathbf{t}_d$ denotes its timestamp.

Our UMPN model updates the vertex and edge representations iteratively over time. We denote such updates as $\hat{\mathcal{G}} = f(\mathcal{G}, \Theta)$, where $\Theta$ denotes the learned weights controlling its behavior. The graph $\mathcal{G}$ contains the initial representations, while the updated ones by UMPN are in $\hat{\mathcal{G}}$.

**Edge Updates.** The edge representations $\mathbf{e}$ are updated first. Each $\mathbf{e}$ is concatenated with the representations $\mathbf{v}$ of the vertices it connects. This is then fed to a two-layer MLP with layer normalization (Ba et al., 2016) and ReLU activation (Fukushima, 1975) between them. In what follows, all MLP models introduced share this two-layer architecture unless otherwise stated.

The output of the MLP is added to the input edge representation, as in a Resnet (Szegedy et al., 2017). Formally, for an edge $e_{ij}$ connecting the vertices $v_i$ and $v_j$, the updated edge representation $\mathbf{e}_{ij}$ can be written as

$$\tilde{\mathbf{e}}_{ij} = \mathbf{e}_{ij} + \mathrm{MLP}_{\theta^e}([\mathbf{e}_{ij} | \mathbf{v}_i | \mathbf{v}_j]) \, , \tag{2}$$

where $\mathrm{MLP}_{\theta^e}$ is a two-layer MLP parameterized by weights $\theta^e$. Since we have three kinds of edges with different semantic meanings, we use separate MLPs parameterized by $\Theta^{e^v}$, $\Theta^{e^t}$, and $\Theta^{e^c}$, one for each kind.

**Vertex Updates.** Vertex representations are updated in two steps. First, given a vertex $v_i$, the representations of its connected edges are transformed into messages. For each edge $e_{ij}$, the message $\mathbf{m}_{ij}$ is computed as

$$\mathbf{m}_{ij} = \mathrm{MLP}_{\theta^m}([\tilde{\mathbf{e}}_{ij} | \mathbf{v}_i | \mathbf{v}_j]) \, , \tag{3}$$

As for the edge update, we use three different MLPs parameterized by $\Theta^{m^v}$, $\Theta^{m^t}$, and $\Theta^{m^c}$ to generate the messages. Then, each message type—derived from its edge type—is aggregated by taking their vertex $v_i$

$$\tilde{\mathbf{m}}_i^v = \frac{1}{|\mathcal{N}^v(i)|} \sum_{j \in \mathcal{N}^v(i)} \mathbf{m}_{ij}^v \, , \tag{4}$$

where $\mathcal{N}^v(i)$ is the set of neighboring vertices connected to $v_i$ via view edges. Therefore, $\tilde{\mathbf{m}}_i^v \in \mathbb{R}^d$ contains the aggregated information from the view messages/edges. $\tilde{\mathbf{m}}_i^t$ and $\tilde{\mathbf{m}}_i^c$ are computed similarly from temporal and contextual connectivities.

Finally, the updated vertex representation is taken to be

$$\tilde{\mathbf{v}}_i = \mathbf{v}_i + \tilde{\mathbf{m}}_i^v + \tilde{\mathbf{m}}_i^t + \tilde{\mathbf{m}}_i^c \ . \tag{5}$$

We hereafter denote the set of MLP parameters used to update the vertex and edge representations as $\Theta_u = (\Theta^{e^v}, \Theta^{e^t}, \Theta^{e^c}, \Theta^{m^v}, \Theta^{m^t}, \Theta^{m^c})$.

**Weight Assignment.** After updating the representations, the final step is to compute classification scores for each vertex and each temporal or inter-view edge. We write the probability that vertices $v_i$ and $v_j$ – connected by edge $e_{ij}$ – belong to the same person as

$$P(\mathbf{e}_{ij}) = \sigma(\mathrm{MLP}_{\theta_{p^e}}([\mathbf{e}_{ij}|\mathbf{v}_i|\mathbf{v}_j])) \tag{6}$$

The probability that vertex $v_i$ represents a true positive detection is taken to be

$$P(\mathbf{v}_i) = \sigma(\mathrm{MLP}_{\theta_{p^v}}(\mathbf{v}_i)) \ , \tag{7}$$

where $\sigma$ is the sigmoid function. The classification head is parameterized by $\Theta_{cls} = (\Theta_{p^{e^v}}, \Theta_{p^{e^t}}, \Theta_{p^v})$.

## 3.3 Training

Given an input graph $\mathcal{G}$, one forward pass through the proposed network yields $P(\hat{\mathcal{E}}), P(\hat{\mathcal{V}}), \hat{\mathcal{G}} = \mathrm{UMPN}(\mathcal{G}, \Theta_{enc}, \Theta_u, \Theta_{cls})$. Here, $P(\hat{\mathcal{E}})$ represents the probabilities of all the temporal and inter-view edges, $P(\hat{\mathcal{V}})$ the vertex probabilities, and $\hat{\mathcal{G}}$ the graph with updated vertex and edge representations, as depicted in Fig. 2.

**Loss Function.** We train our UMPN to learn the parameters $\Theta_{enc}$, $\Theta_u$ and $\Theta_{cls}$ so that the view and temporal edges, along the vertices, of $\mathcal{G}$ are correctly classified. To this end, we use ground-truth edge/vertex labels. Each edge $e_{i,j}$ and vertex $v_i$ is annotated with its ground-truth binary class label $e_{i,j}^*$ and $v_i^*$, respectively. We minimize a multi-task loss based on focal losses (Lin et al., 2017). We write it as

$$\mathcal{L} = \mathcal{L}_{\mathrm{focal}}(e^{v*}, P(\mathbf{e}^v)) + \mathcal{L}_{\mathrm{focal}}(e^{t*}, P(\mathbf{e}^t)), + \mathcal{L}_{\mathrm{focal}}(v^*, P(\mathbf{v})) \ ,$$
$$\mathcal{L}_{\mathrm{focal}}(y, p) = -y(1-p)^\gamma \log(p) - (1-y)p^\gamma \log(1-p) \ , \tag{8}$$

where $\mathcal{L}_{\mathrm{focal}}$ is the focal loss for a binary classification task, $y \in \{0, 1\}$ is the ground truth label, $p \in [0, 1]$ is the prediction, and $\gamma$ is a focusing parameter. This provides a robust loss that down-weights the contribution of easy examples.

**Dynamic Learning.** Recall from Section 3.1, that we recursively build spatio-temporal graphs that cover temporal windows of length $W$. Each time it is updated, we compute the supervised training loss of Eq. (8). We then move the temporal window by one time step, discarding the oldest nodes and edges from time step $t - W + 1$ and adding new ones from time step $t + 1$. In the resulting graph, the node and edge representations between $t - W + 2$ and $t + 1$ are the results of the previous iteration and already encode spatio-temporal context. We iterate this process until the end of the sequence. As detailed in Section 5.1, the loss is accumulated over several frames before being backpropagated to update the model parameters.

By exposing the model to subsequences of dynamically expanding graphs during training, the model learns to smoothly accumulate information over long sequences, which is further reinforced by delaying the backward pass in order to accumulate gradients over multiple time-steps. This strategy allows the model to learn long-term dependencies and to generalize better. Additionally, it avoids relying on heuristics or ad hoc aggregations when transitioning between windows. This provides a principled approach for graph networks to handle indefinite-length sequential data, as needed for online inference.

| Dataset | #Cam. | FPS | #Frames | #IDs | Geom. | Avg. Clip Length | #Clips |
|---|---|---|---|---|---|---|---|
| MOT17 [41] | 1 | 14-30 | 11k | 1.3k | None | 33s | 14 |
| MOT20 [16] | 1 | 25 | 13k | 3.5k | None | 67s | 8 |
| PETS2009 [20] | 7 | 7 | 795 | 19 | Plane | 22s | 4 |
| WILDTRACK [12] | 7 | 2 | 2.8k | 313 | Plane | 3min20s | 1 |
| SCOUT (Ours) | 25 | 10 | 564k | 3k | Mesh | 20min | 2 |

Table 1: Our dataset compared to other pedestrian datasets.

### 3.4 From Probabilistic Graph to Trajectories

To obtain discreet trajectories from the probabilities obtained from our model, we rely on a fast online greedy algorithm compatible with both single and multiple camera setups. Its simplicity is made possible by the quality and consistency of computed probabilities. In the supplementary material, we demonstrate that it consistently finds near-optimal solutions from the generated probabilistic graphs.

The algorithm first filters out low-confidence detections and edges. Then it converts the graph into an adjacency list sorted by confidence. Finally it processes the detections sequentially and merges them into trajectories while ensuring temporal and spatial consistency. The only constraint is that a trajectory cannot contain more than one detection from the same camera at the same time-step. The pseudocode of our greedy approach is available in the supplementary material. As in the training phase, the algorithm processes the graph in an online fashion within a rolling temporal window of size $W$. This is done by running it on a sub-graph containing all the vertices and the edges of the last W frames at each timestep.

## 4 Datasets

We evaluate our approach on three publicly available datasets, WILDTRACK (Chavdarova et al., 2018), MOT17 (Milan et al., 2016) and MOT20 (Dendorfer et al., 2021). We use the MOT17 train/val split from (Zhou et al., 2020). As these datasets are relatively small and do not provide much information about the structure of the surrounding scenes, we created our own larger-scale one that covers both single- and multi-view scenarios. As shown in Table 1, it features 25 calibrated cameras and 564,000 frames, which is much more than the other two. It includes camera calibration and 3D scene geometry for both single and multi-view evaluation. We dub it SCOUT for Scene Context and Occlusion Understanding. In the remainder of this section, we explain how it was created.

### 4.1 Data Acquisition

In Fig. 3, we show the placement and the region of interest for each camera.

**Hardware.** The dataset was collected using 25 cameras. Each camera setup included a Raspberry Pi 4 and a Camera Module 3 with a wide-angle lens. The videos were recorded at 30 FPS with a resolution of 1920x1080. In parallel to the video files, timestamps for each frame were recorded. The synchronization between cameras was achieved using the NTP protocol. The cameras were mounted on existing scene structures such as pillars, beams and roofs at an average height of 2.8 meters.

**Sequences.** The collection was run for two windows of 1h each. From each window, sequences of 20 minutes were selected and sub-sampled at 10 Hz frame rate, which resulted in 12000 synchronized frames per sequence and per camera. Due to faults during the acquisition campaign, one of the sequences features only 22 cameras while the other one features 25.

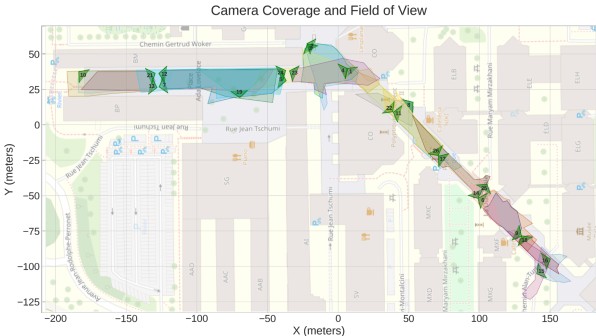

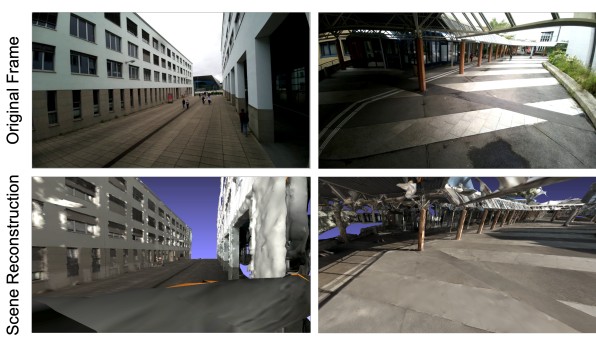

Figure 3: **Scene structure and camera placement.** Overlapping cameras cover a 450m path. Green triangles indicate camera positions and fields of view, while polygons mark visible ground areas.

Figure 4: **Scene reconstruction.** The top row displays frames captured by two distinct cameras and the bottom row presents their reconstructed textured scene meshes.

### 4.2 Scene Reconstruction

We provide the scene reconstruction as a textured mesh in a metric coordinate system. We carried a 360 degree camera equipped with a GPS receiver through the entire scene, overlapping with the fields of view of all the cameras. The mesh was then built using OpenSFM (Mapillary, 2021). Fig. 4 illustrates sample scene reconstructions.

**Calibration.** The cameras are calibrated using a pinhole model (Tsai, 1987). While the intrinsic parameters are computed using a calibration target, the extrinsics are obtained using the scene reconstruction as an intermediate step. More specifically, we run a PnP algorithm (Lepetit et al., 2009) on the dense matches between the 360 degree footage and the individual camera frames to estimate the camera poses.

### 4.3 Annotation

We rely on a semi-automated process to annotate our dataset. First, we extract monocular trajectories using a pre-trained version of ByteTrack (Zhang et al., 2022). Next, we convert the detected bounding boxes to 3D bounding cylinders using the calibration and the reconstructed mesh. We then merge 3D trajectories that are close to each other, representing the same object viewed by different cameras. Finally, we developed a tool to visualize and clean up the annotations directly in 3D, displaying all overlapping views at once. Manual cleanup is performed at 1 FPS, and once completed, the annotations are linearly interpolated to 10 FPS.

## 5 Experiments

We evaluate our approach on the three datasets of Section 4.

### 5.1 Metrics and Implementation Details

We evaluate tracking performance using standard metrics from the literature. For both single-view and multi-view scenarios, we report CLEAR MOT metrics (Kasturi et al., 2009) - MOTA (tracking accuracy) and MOTP (localization precision), along with IDF1 (Ristani et al., 2016) for identity preservation. For single-view tracking, we additionally report HOTA (Luiten et al., 2020). Multi-view metrics are computed in 3D using *py-motmetrics*, while single-view metrics use *trackeval* (Luiten & Hoffhues, 2020) in 2D space.

Our dynamic training approach processes frames sequentially rather than in mini-batches. We split sequences into chunks of 40 frames, initializing an empty graph at the start of each chunk. For each frame, we expand

| | Method | MOTA ↑ | MOTP ↑ | IDF1 ↑ |
|---|---|---|---|---|
| WILDTRACK | KSP-DO [12] | 69.6 | 61.5 | 73.2 |
| | KSP-DO-ptrack [12] | 72.2 | 60.3 | 78.4 |
| | GLMB-YOLOv3 [44] | 69.7 | 73.2 | 74.3 |
| | GLMB-DO [44] | 70.1 | 63.1 | 72.5 |
| | DMCT [65] | 72.8 | 79.1 | 77.8 |
| | DMCT Stack [65] | 74.6 | 78.9 | 81.9 |
| | ReST [13] | 84.9 | 84.1 | 86.7 |
| | EarlyBird [58] | 89.5 | 86.6 | 92.3 |
| | MVFlow [17] | 91.3 | - | 93.5 |
| | TrackTacular [59] | 91.8 | 85.4 | 95.3 |
| | UMPN (Ours) | **93.9** | **86.9** | **96.3** |

Table 2: **Comparative results on WILDTRACK.** Our approach achieves significantly better results than the prior state-of-the-art.

| | Method | MOTA↑ | IDF1↑ | HOTA↑ |
|---|---|---|---|---|
| | **MOT17** | | | |
| Val | SORT [8] | 75.2 | 75.2 | 65.3 |
| | ByteTrack [68] | 76.8 | **76.9** | 66.2 |
| | UMPN (Ours) | **82.2** | 75.7 | **66.8** |
| Test | CenterTrack [70] | 67.8 | 64.7 | 52.2 |
| | TransTrack [54] | 75.2 | 63.5 | 54.1 |
| | DiffMOT [38] | 79.8 | **79.3** | **64.5** |
| | GeneralTrack [45] | 80.6 | 78.3 | 64.0 |
| | ByteTrack [68] | 80.3 | 77.3 | 63.1 |
| | UMPN (Ours) | **82.2** | 75.1 | 62.5 |
| | **MOT20** | | | |
| Val | ByteTrack [68] | 63.2 | 74.9 | 59.7 |
| | UMPN (Ours) | **71.6** | **76.5** | **60.3** |
| Test | TransTrack [54] | 65.0 | 59.4 | 48.5 |
| | GeneralTrack [45] | 77.2 | 74.0 | 61.4 |
| | ByteTrack [68] | **77.8** | **75.2** | 61.3 |
| | UMPN (Ours) | 77.7 | **75.2** | **62.0** |

Table 3: **Results on MOT17 and MOT20** using the MOTChallenge (Milan et al., 2016) private evaluation protocol. Best results in **bold**.

the graph and perform a forward pass through our UMPN. The loss is computed and accumulated over the chunk, with a backward pass and parameter update occurring at the chunk's end.

We maintain a rolling window of size $W = 10$ frames - once the graph reaches this size, we prune the oldest frame's nodes and edges when adding new ones, as described in Section 3.1. When pruning, we extract edge and vertex scores which are fed into our trajectory extraction algorithm. We discard vertices and edges with probabilities below thresholds $\tau_n = 0.5$ and $\tau_e = 0.5$ respectively.

To optimize graph construction, we enforce physical constraints: temporal edges are pruned if they imply speeds above 3 m/s, while view edges are limited to a 1m maximum distance to handle detection and calibration noise. We restrict the temporal distance between connected vertices to 4 frames during training and 6 frames at inference. For regularization, we randomly prune 10% of edges, mask 5% of vertex features, and apply dropout with rate 0.1 to all MLPs.

| Method | Metrics | | | |
|---|---|---|---|---|
| | MOTA↑ | MOTP↑ | IDF1↑ | HOTA↑ |
| **Single-View** | | | | |
| SORT [8] | 22.7 | - | 35.8 | 33.8 |
| ByteTrack [68] | 35.7 | - | 43.9 | 49.4 |
| UMPN (Ours) | 69.8 | - | 62.1 | 63.3 |
| UMPN (Ours) + SP | **73.6** | - | **63.3** | **64.8** |
| **Multi-View** | | | | |
| ByteTrack MV | 62.1 | 85.1 | 62.3 | |
| UMPN (Ours) | 79.0 | 86.3 | 83.2 | - |
| UMPN (Ours) + SP | **81.3** | **87.0** | **83.3** | - |

*(SCOUT)*

Table 4: **Comparative results on our Scout dataset.** Results are grouped by tracking scenario (single-view vs multi-view). Best results in each category shown in **bold**. SP stands for Scene Prior. See last paragraph of Section 3.1 or Section A.1 for more information

## 5.2 Benchmarking

**WILDTRACK.** We use the WILDTRACK dataset to evaluate our method in a multi-view setting. To represent detection we use their 2D bounding box coordinates, world coordinates, view index and time-step. World coordinates are derived from the 2D bounding box coordinates and the camera calibration using the flat ground plane assumption (Chavdarova et al., 2018). The encoding dimension of those attributes yield vertex feature dimension of $s = 704$. We use a pre-trained people detector (Engilberge et al., 2023) comparable to previous work (Engilberge et al., 2023; Teepe et al., 2024).

We report our results in Table 2. Our method significantly outperforms previous state-of-the-art (SOTA) methods boosting MOTA by 2.6% to 93.9. In terms of identity preservation (IDF1), it achieves a nearly 3% improvement over the SOTA reaching 96.3.

**MOT17 and MOT20.** For single-view evaluation, we use the MOT17 and MOT20 datasets. Each detection is represented by a feature vector containing its 2D bounding box coordinates, timestamp, and confidence score, yielding a vertex feature dimension of $s = 640$ for MOT17 and $s = 300$ for MOT20. As in (Zhang et al., 2022; Sun et al., 2020; Qin et al., 2024), we use a pre-trained YOLOX people detector (Ge et al., 2021). Since this paper focuses on linking detections, rather than improving them, this makes for a fair comparison. As can be seen in Tab 3, we outperform ByteTrack on validation on both MOT20 and MOT17. On the private test set, we surpass it in terms of MOTA on MOT17 and match its performance on MOT20, though our IDF1 score lags behind on MOT17. We attribute this performance gap to memory constraints during training, limiting our temporal connections to 6 frames, while ByteTrack can handle gaps of up to 30 frames.

For completeness, we also report the numbers for more recent methods. On MOT17, DiffMOT (Lv et al., 2024) and GeneralTrack (Qin et al., 2024) outperform us on IDF1 and HOTA, but we outperform them on MOTA. This can be explained by the fact that they rely heavily on appearance information to associate detections, which we do not use. On the other hand, appearance seems to be detrimental on the crowded MOT20 where we outperform GeneralTrack on all metrics. In any case, the fact that our numbers are close without using appearance is encouraging and points to an obvious direction for future research: Exploiting appearance features while being able to handle single and multi-view information.

**SCOUT.** We evaluate on our SCOUT dataset, which provides both single and multi-view tracking scenarios. We use the same detection attributes as in WILDTRACK but we reduce the feature dimensions to $s = 224$ in the multi-view case to fit the larger graph in memory. We split the dataset into two equal parts: the first is used for training, and the second for evaluation. We report our comparative results in Table 4. Overall the metrics are significantly lower than on MOT17 and WILDTRACK because the SCOUT dataset is more demanding than previous ones. Nevertheless, for single-view evaluation, we significantly outperform

| | Configuration | MOTA↑ | IDF1↑ | HOTA↑ |
|---|---|---|---|---|
| MOT17 | w/o context edges | 75.5 | 69.5 | 62.4 |
| | Min Det. Conf 0.3 | 77.5 | 72.3 | 64.6 |
| | Min Det. Conf 0.5 | 75.1 | 69.0 | 61.6 |
| | Max Temp. dist. 1 | 77.34 | 59.4 | 56.5 |
| | Max Temp. dist. 4 | 81.4 | 72.5 | 64.9 |
| | Max Temp. dist. 8 | 81.6 | 74.9 | 66.1 |
| | Full model | **82.2** | **75.7** | **66.8** |

Table 5: **Graph construction ablation on MOT17.** Context edges and longer temporal connections improve tracking performance, while lower detection confidence thresholds help capture more targets.

| | Attributes Used | | | | Metrics | | |
|---|---|---|---|---|---|---|---|
| | Box | World | View | Time | MOTA↑ | MOTP↑ | IDF1↑ |
| WILDTRACK | ✓ | | | | 93.5 | 86.1 | 92.9 |
| | | ✓ | | | 90.9 | 85.0 | 94.1 |
| | ✓ | ✓ | | | 91.7 | 86.3 | 95.0 |
| | ✓ | | ✓ | | 92.6 | 86.2 | 93.6 |
| | ✓ | ✓ | ✓ | | 93.6 | 86.3 | 94.7 |
| | Training Configurations - All Attributes | | | | | | |
| | $s = 356$ | | | | 91.4 | 85.2 | 95.8 |
| | $s = 176$ | | | | 90.9 | 86.7 | 92.8 |
| | $s = 88$ | | | | 80.4 | 81.6 | 88.0 |
| | $W = 5$ | | | | 92.4 | **87.3** | 93.7 |
| | $W = 2$ | | | | 88.3 | 85.2 | 89.6 |
| | w/o edge dropout | | | | 92.3 | 86.5 | 91.2 |
| | w/o context edges | | | | 85.5 | 80.4 | 90.4 |
| | w/o delayed backward | | | | 43.8 | 72.4 | 34.0 |
| | Full model | | | | **93.9** | 86.9 | **96.3** |

Table 6: **Ablation studies on WILDTRACK.** Using all detection attributes and delayed backpropagation optimizes performance, while edge pruning and larger embeddings further enhance results.

SORT (Bewley et al., 2016) and ByteTrack (Zhang et al., 2022) and incorporating scene information gives us an additional boost.

In the multi-view setting, existing methods (Engilberge et al., 2023; Teepe et al., 2024;) could not be run because their memory requirements were too high. To nevertheless provide a multi-view baseline, we modified ByteTrack (Zhang et al., 2022) to cluster detections across views and average their world coordinates per cluster before tracking. Instead of 2D IoU, we used 3D IoU between cuboids for similarity computation. Both our models, with and without environmental knowledge, outperform the baseline, with the former achieving slightly better results.

### 5.3 Further Analysis

To refine our understanding of our model's behavior, we conduct several ablation studies. First we analyze the effect of detection attributes used to initialize the vertex features. Then we study the impact of the graph construction method. Finally we evaluate the effect of different training strategies.

**Detection attribute ablation.** On the WILDTRACK dataset, we test the effect of individual detection attributes used to initialize the vertex features. We test bounding box coordinate, world coordinate, view index, timestep and their combination. You can find results in the top part of Table 6. When using single attributes we see that the bounding box coordinate alone outperforms world coordinate this is consistent with observations from previous work (Bergmann et al., 2019). The combination of the two improves IDF1. Interestingly, combining bbox with view index is beneficial for IDF1. Using the timestep improves performance across all metrics.

**Graph Construction Ablation.** Since our model heavily relies on the graph structure, we analyze how different graph construction choices affect tracking performance on MOT17 (Table 5).

Our experiments show that contextual edges are essential for information flow, as removing them significantly lowers performance. With a default confidence threshold of 0.1, our model best leverages weak signals, using low-confidence detections effectively while filtering out false positives. Testing higher thresholds (0.3 and 0.5) degrades performance, as more weak signals are removed.

We analyze the impact of temporal edge pruning on IDF1 score. At inference time, limiting edges to 1 frame reduces performance, as objects cannot be recovered after brief occlusions. Extending to 4 frames

| Model | Runtime | | | | Speed |
| | Detection | Graph | Trajectory | Total | FPS |
|---|---|---|---|---|---|
| SORT [8] | 17ms | - | 12ms | 29ms | 34 |
| ByteTrack [68] | 17ms | - | 13ms | 30ms | 33 |
| UMPN (Ours) | 17ms | 10ms | 17ms | 44ms | 23 |

Table 7: **Runtime analysis.** Runtime for a single frame and for the different components in our tracking pipeline compared to ByteTrack and SORT. All times measured on an NVIDIA A100 GPU.

(the training value) improves results, with the best performance seen at 6 frames. Increasing to 8 frames, however, proves ineffective, likely due to the training distribution mismatch.

**Training Strategy Ablation.** We evaluate key training elements on WILDTRACK, with results in Table 6. Higher embedding dimensionality ($s$) improves performance up to $s = 704$, though GPU memory limits further exploration. Larger temporal windows ($W$) boost tracking quality, with $W = 10$ improving MOTA by 1.5% compared to $W = 5$. Random edge dropout during training enhances robustness, while our delayed backpropagation strategy proves crucial - replacing it with standard backpropagation causes performance to drop by more than 50% on all metrics.

**Timing results.** While our method is a learned approach, it maintains competitive computational efficiency compared to handcrafted methods. Timing information is provided in Table 7. The runtime is in the range of other state-of-the-art methods and of the YOLOx detection network. Making it suitable for real-time applications with an average frame rate of 23 FPS.

**Limitations.** Our dynamic training strategy delays back-propagation over longer sequences. Thus, the memory required to store the gradients is higher than for standard training. This has been a bottleneck for certain parameters of our model such as the window size or the maximum temporal distance between connected vertices. This in turn limits the maximum number of frames an object can be missing before the model can no longer recover its trajectory. However this memory limitation is only present at training time and not at inference time. Hence at inference time we can connect objects missing for longer periods. This improves identity preservation but is suboptimal as it creates a mismatch between training and inference regimes.

**Broader Impact.** Our multi-view multi-object tracking model advances tracking capabilities for applications in security, autonomous systems, and smart environments. These benefits, however, come with privacy and ethical considerations, particularly in surveillance contexts.

To address these, our data collection protocol received ethical approval, and participants were fully informed of the process, with clear options to avoid data capture or request removal. This ensures respect for privacy and individual rights. By releasing our dataset, we encourage further research aligned with ethical standards, promoting responsible and privacy-conscious applications of tracking technologies.

## 6  Conclusion

We have introduced a unified, fully differentiable approach to multi-people tracking that leverages a dynamic spatiotemporal graph representation. Our model efficiently aggregates spatial, contextual, and temporal information across sequences through learned message passing on this graph structure. The flexibility of our approach enables robust tracking in both single-view and multi-view scenarios, as demonstrated by state-of-the-art results on public benchmarks like WILDTRACK and our newly created SCOUT dataset. While we focused on people tracking, our framework is general and could be extended to track diverse object categories in future work with minimal modifications.

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

# A   Appendix

## A.1   Graph with Scene Structure

**Camera Vertices and Edges.**   In order to model scene structure, we add vertices to the graph corresponding to cameras. If the camera is stationary, we add a single vertex to the graph. If the camera is moving, we add a vertex for each frame in the sequence. We connect each camera vertex to all detections vertices in the graph. The edges between camera vertices are associated with features that encode visibility, origin of detection and distance to the camera. We denote the camera vertices as $v^c$ and their representation as $\mathbf{v}^c$. We denote the edges involving camera vertices as $e^{cd}$ and their representation as $\mathbf{e}^{cd}$.

**Representation of camera vertices.**   In similar fashion to the detection vertices, the camera vertices are initialized using a two-layer MLP to encode their attributes. The camera vertices encode the camera's world coordinates $\mathbf{w}_c$ along with its intrinsic parameters $\mathbf{K}$ and extrinsic parameters $[\mathbf{R}|\mathbf{t}]$.

Edges between cameras and detections encode three key attributes: the Euclidean distance between the camera and detection in world coordinates, a binary indicator of whether the detection originated from that camera, and a visibility flag indicating if the detection has a clear line of sight to the camera.

$$\mathbf{e}^{cd^0} = \left[\mathrm{MLP}\left(d(\mathbf{w}_c, \mathbf{w}_d)\right) | \mathrm{MLP}\left(\mathbf{b}_{\mathrm{src}}\right) | \mathrm{MLP}\left(\mathbf{b}_{\mathrm{vis}}\right)\right],$$

$$\mathbf{v}^{c^0} = \left[\mathrm{MLP}\left(\mathbf{w}_c\right) | \mathrm{MLP}\left(\mathbf{K}\right) | \mathrm{MLP}\left([\mathbf{R}|\mathbf{t}]\right)\right] \tag{9}$$

where $d(\cdot, \cdot)$ computes the Euclidean distance, $\mathbf{b}_{\mathrm{src}}$ indicates if the detection originated from the camera, and $\mathbf{b}_{\mathrm{vis}}$ indicates if the detection is visible to the camera. The visibility flag is computed using the detailed mesh of the scene by checking if the ray from the camera to the detection intersects the mesh before reaching the detection.

**Camera edge updates.** Similar to detection edges, camera edges are updated at each time step using residual updates:

$$\tilde{\mathbf{e}}_{ij}^{cd} = \mathbf{e}_{ij}^{cd} + \mathrm{MLP}_{\theta^e}\left(\left[\mathbf{e}_{ij}^{cd} | \mathbf{v}_i^c | \mathbf{v}_j\right]\right), \tag{10}$$

**Camera vertex updates.** Camera vertices are updated in two steps. First, given a camera vertex $v_i^c$, messages are computed from each connected detection vertex $v_j$ as:

$$\mathbf{m}_{ij}^{cd} = \mathrm{MLP}_{\theta^m}\left(\left[\mathbf{e}_{ij}^{cd} | \mathbf{v}_i^c | \mathbf{v}_j\right]\right),$$

$$\tilde{\mathbf{m}}_i^{cd} = \frac{1}{|\mathcal{N}^{cd}(i)|} \sum_{j \in \mathcal{N}^{cd}(i)} \mathbf{m}_{ij}^{cd}, \tag{11}$$

where $\mathcal{N}^{cd}(i)$ denotes the set of detection vertices connected to camera vertex $i$. Finally, the updated camera vertex representation is:

$$\tilde{\mathbf{v}}_i^c = \mathbf{v}_i^c + \tilde{\mathbf{m}}_i^{cd}. \tag{12}$$

**Detection vertex updates.** When camera vertices are added to the graph, the detection vertices are updated to account for the new connections. Messages from connected camera vertices are computed as:

$$\tilde{\mathbf{m}}_i^{dc} = \frac{1}{|\mathcal{N}^{dc}(i)|} \sum_{j \in \mathcal{N}^{dc}(i)} \mathbf{m}_{ji}^{cd}, \tag{13}$$

where $\mathcal{N}^{dc}(i)$ denotes the set of camera vertices connected to detection vertex $i$. The updated detection vertex representation combines the original messages with the camera messages:

$$\tilde{\mathbf{v}}_i = \mathbf{v}_i + \tilde{\mathbf{m}}_i^v + \tilde{\mathbf{m}}_i^t + \tilde{\mathbf{m}}_i^c + \tilde{\mathbf{m}}_i^{dc}. \tag{14}$$

## A.2   3D Projection

Let $(x_1, y_1)$ and $(x_2, y_2)$ indicate the top left and bottom right pixel coordinates of the object bounding box, respectively. Given the homogeneous pixel coordinates $\tilde{\mathbf{u}}_i = [(x_1 + x_2)/2, y_2, 1]^\top$ of the object's base point in the view from camera $i$, along with the intrinsic matrix $\mathbf{K}_i$, rotation matrix $\mathbf{R}_i$, and translation vector $\mathbf{t}_i$ of camera $i$, we compute an intermediate ray direction $\mathbf{l} = \mathbf{R}_i^\top \cdot \mathbf{K}_i^{-1} \cdot \tilde{\mathbf{u}}_i$ and the camera's position in world coordinates $\mathbf{c}_i = -\mathbf{R}_i^\top \cdot \mathbf{t}_i$. The ray from the optical center of camera $i$ through the pixel's image coordinate in 3D space is given by $\mathbf{Q}_i(\lambda) = \mathbf{c}_i + \lambda \cdot \mathbf{l}_i$. To find the world point $\mathbf{p}$, we compute the intersection of this ray with the scene mesh using ray-triangle intersection tests. For each triangle in the mesh, we solve:

$$\mathbf{c}_i + \lambda \cdot \mathbf{l}_i = \mathbf{v}_0 + \beta(\mathbf{v}_1 - \mathbf{v}_0) + \gamma(\mathbf{v}_2 - \mathbf{v}_0) \tag{15}$$

where $\mathbf{v}_0$, $\mathbf{v}_1$, and $\mathbf{v}_2$ are the vertices of the triangle, and $\beta$ and $\gamma$ are barycentric coordinates. The intersection point with the smallest positive $\lambda$ where $\beta + \gamma \leq 1$ and $\beta, \gamma \geq 0$ gives us the world point $\mathbf{p}$ where the ray first hits the scene geometry.

When no scene mesh is available, as in datasets like WILDTRACK, we instead assume a flat ground plane at z=0. In this case, we can directly compute the intersection point by solving for $\lambda$ where the ray meets the ground plane:

$$\mathbf{c}_i + \lambda \cdot \mathbf{l}_i = \begin{bmatrix} x \\ y \\ 0 \end{bmatrix} \tag{16}$$

This simplifies to solving the linear equation:

$$\lambda = -\frac{\mathbf{c}_i[2]}{\mathbf{l}_i[2]} \tag{17}$$

where $\mathbf{c}_i[2]$ and $\mathbf{l}_i[2]$ are the z-components of the camera center and ray direction respectively. The final world point is then:

$$\mathbf{p} = \mathbf{c}_i + \lambda \cdot \mathbf{l}_i \tag{18}$$

This approximation works well for scenes with relatively flat ground surfaces, though it may introduce some error in areas where the actual ground deviates significantly from the z=0 plane.

### A.3  From Probabilistic Graph to Trajectories

The pseudocode of our greedy approach is presented in Algorithm 1. Our algorithm extracts trajectories from the probabilistic graph by merging detections that are connected by high-confidence edges while maintaining spatio-temporal consistency. The key constraint, enforced by the ValidMerge function, ensures that a trajectory cannot contain multiple detections from the same timestamp and camera view, as this would violate physical constraints. The algorithm processes both temporal edges (connecting detections across time) and view edges (connecting simultaneous detections across cameras) to build globally consistent trajectories. By using a disjoint set data structure, we can efficiently merge compatible detections while maintaining the consistency constraints.

### A.4  SCOUT Additional Results

The memory constraints of previous methods prevent a direct comparison with our current approach on the complete SCOUT dataset. Therefore, we evaluate our method using a subset of 8 cameras over a single 20-minute sequence. The subset is selected to cover the most challenging scenarios present in the dataset (e.g., occlusions, crowded scenes, and distant detections). Half of the sequence is used for training and the other half for testing. The results are presented in Table 8, with MVFlow (17) serving as the baseline. The proposed method outperforms MVFlow by 7.6 points on MOTA and 2 points on IDF1. This improvement can be attributed to the non-flat nature of the scene (staircase), which cannot be modeled by the ground-plane homography used in MVFlow. Similar to Table 4, it also outperforms the ByteTrack-MV baseline.

### A.5  Memory Consumption

We study the GPU memory consumption of our model at training and inference time for different maximum temporal distances between connected vertices. Results can be found in Table 9. The memory consumption at training grows significantly with the increase of the maximum temporal distance. This is expected as the delayed backpropagation requires to store more gradients for any additional eges. At inference time the memory consumption is significantly lower and only grow marginally allowing for edges modeling much longer temporal distances.

---

**Algorithm 1** Multi-View Trajectory Extraction

---

**Require:** Input graph $\mathcal{G} = (\mathcal{V}, \mathcal{E})$ with vertices $\mathcal{V}$ and edges $\mathcal{E}$. Probabilities for vertices $P(v)$ and edges $P(e)$. Edge and node confidence thresholds $\tau_e$ and $\tau_n$

**Ensure:** Set of consistent trajectories $\mathcal{T}$

1:   $\mathcal{V}_{\text{valid}} \leftarrow \{v \in \mathcal{V} \mid P(v) > \tau_n\}$
2:   Initialize adjacency lists $\mathcal{A}[v]$ for all $v \in \mathcal{V}_{\text{valid}}$
3:   **for** $t \in \{\text{temporal}, \text{view}\}$ **do**
4:      $\mathcal{E}_t \leftarrow \{e \in \mathcal{E} \mid P(e) > \tau_e\}$
5:      **for** $(u, v) \in \mathcal{E}_t$ **do**
6:        $\mathcal{A}[u] \leftarrow \mathcal{A}[u] \cup \{v\}$
7:      **end for**
8:   **end for**
9:   Initialize disjoint set structure $\mathcal{DS}$ with trajectory metadata
10:   **for** $u \in \mathcal{V}_{\text{valid}}$ **do**
11:      **for** $v \in \mathcal{A}[u]$ **do**
12:        **if** ValidMerge($\mathcal{DS}, u, v$) **then**
13:          $\mathcal{DS}$.Union($u, v$)
14:        **end if**
15:      **end for**
16:   **end for**
17:   $\mathcal{T} \leftarrow$ ExtractTrajectories($\mathcal{DS}$)
18:   **return** $\mathcal{T}$

---

| Method | Metrics | | |
| --- | --- | --- | --- |
| | MOTA↑ | MOTP↑ | IDF1↑ |
| **Multi-View** | | | |
| ByteTrack MV | 19.0 | 25.7 | 24.3 |
| MVFlow [17] | 15.6 | 22.3 | 24.2 |
| UMPN (Ours) | 18.0 | 25.2 | 24.9 |
| UMPN (Ours) + SP | **23.2** | **26.4** | **26.2** |

Table 8: **Comparative results on a subset of the Scout dataset.** Results are obtained on a subset of 8 cameras over a single sequence. Best results in each category shown in **bold**. SP stands for Scene Prior.

## A.6   Effect of Temporal Distance

We evaluate the impact of the maximum temporal distance between connected vertices at inference time. As shown in Fig. 5, increasing this distance beyond the training value of 4 initially improves tracking metrics, with peaks at distances 5-6. This suggests that allowing longer-range temporal connections helps recover more challenging trajectories. However, performance eventually degrades at larger distances, likely because the model operates outside its training regime. This highlights a limitation of our current approach - while longer temporal connections are beneficial, memory constraints during training prevent us from directly optimizing for them.

## A.7   Optimality of Trajectories Extraction Algorithm

We use a probabilistic objective function to evaluate the optimality of the solutions produced by our algorithm. Following the optimization framework introduced in (6), we compute edge costs as the negative log-likelihood ratios of the edge probabilities. The goal is to minimize the sum of these costs for the edges along the trajectories in a given solution.

We compute an optimality gap by comparing the cost of the solution found by our algorithm and the lower bound for any solution given the same graph. The lower bound is defined as the sum of all negative edge

| Max Distance | 1 | 2 | 3 | 4 | 5 | 6 | 10 | 15 | 20 | 30 |
|---|---|---|---|---|---|---|---|---|---|---|
| **Single-View - MOT17** | | | | | | | | | | |
| Training | 3.6 | 4.4 | 5.4 | 6.4 | 7.4 | 8.4 | 10.1 | 23.64 | - | - |
| Inference | 0.16 | 0.16 | 0.16 | 0.17 | 0.17 | 0.17 | 0.17 | 0.19 | 0.20 | 0.47 |
| **Multi-View - WILDTRACK** | | | | | | | | | | |
| Training | 12.4 | 15.7 | 19.8 | 24.6 | 29.4 | 34.0 | - | - | - | - |
| Inference | 0.22 | 0.23 | 0.25 | 0.27 | 0.29 | 0.31 | 0.38 | 0.44 | 0.7 | 1.19 |

Table 9: Average memory usage (GB) at training and inference time for different maximum temporal distances between connected vertices. Missing values indicate that peak memory usage was higher than 80GB and could not train for a full epoch due to the limitation from the GPU we used.

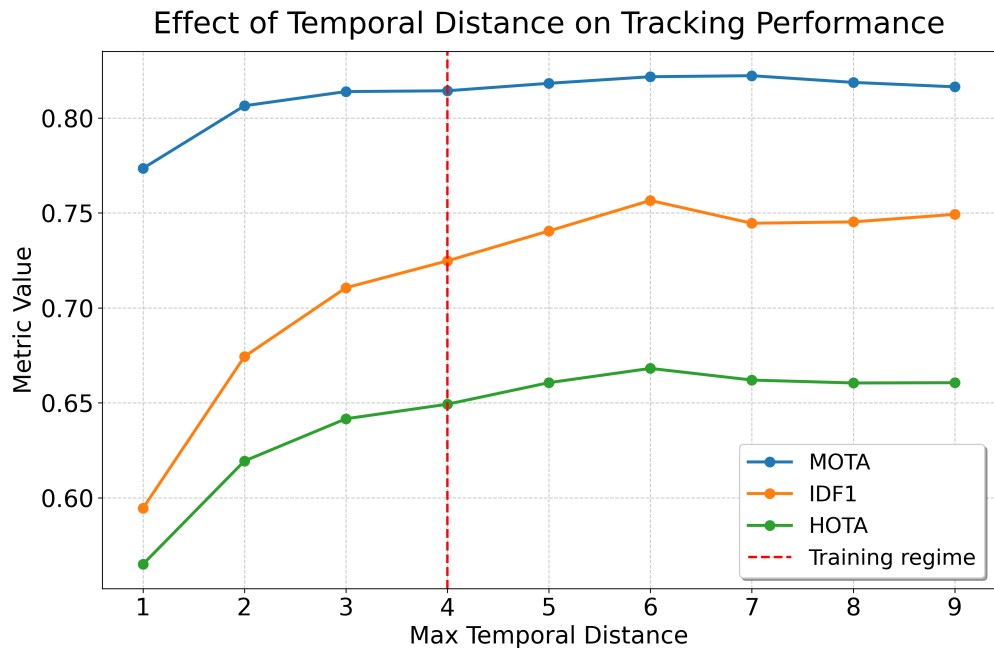

Figure 5: Impact of maximum temporal distance between connected vertices at inference time on tracking metrics. The model was trained with max distance 4. Increasing it initially improves performance but eventually degrades as it diverges from the training regime.

costs in a given graph and, hence, does not account for any spatio-temporal consistency constraints that a feasible solution must satisfy but no solution can have a lower cost.

On WILDTRACK, we obtain an optimality gap of 4.48% for the results in Table 2. Similarly, on MOT17, we obtain a mean optimality gap of 3.75% corresponding to the results in Table 3. This shows that our greedy algorithm is able to find solutions that are close to the lower bound.

In other words, further improvement of the tracking metrics needs to come from better graph predictions.

## A.8 SCOUT Dataset Statistics

The SCOUT dataset comprises 564k frames captured from 25 cameras and includes 3k unique individuals, resulting in a total of 3.8 million 2D annotations. On average, individual trajectories span 600 frames, corresponding to approximately 60 seconds and 80 meters of movement. This is significantly longer than those in previous datasets: MOT sequences typically last less than 30 seconds, while the WILDTRACK sequence extends to 3 minutes.

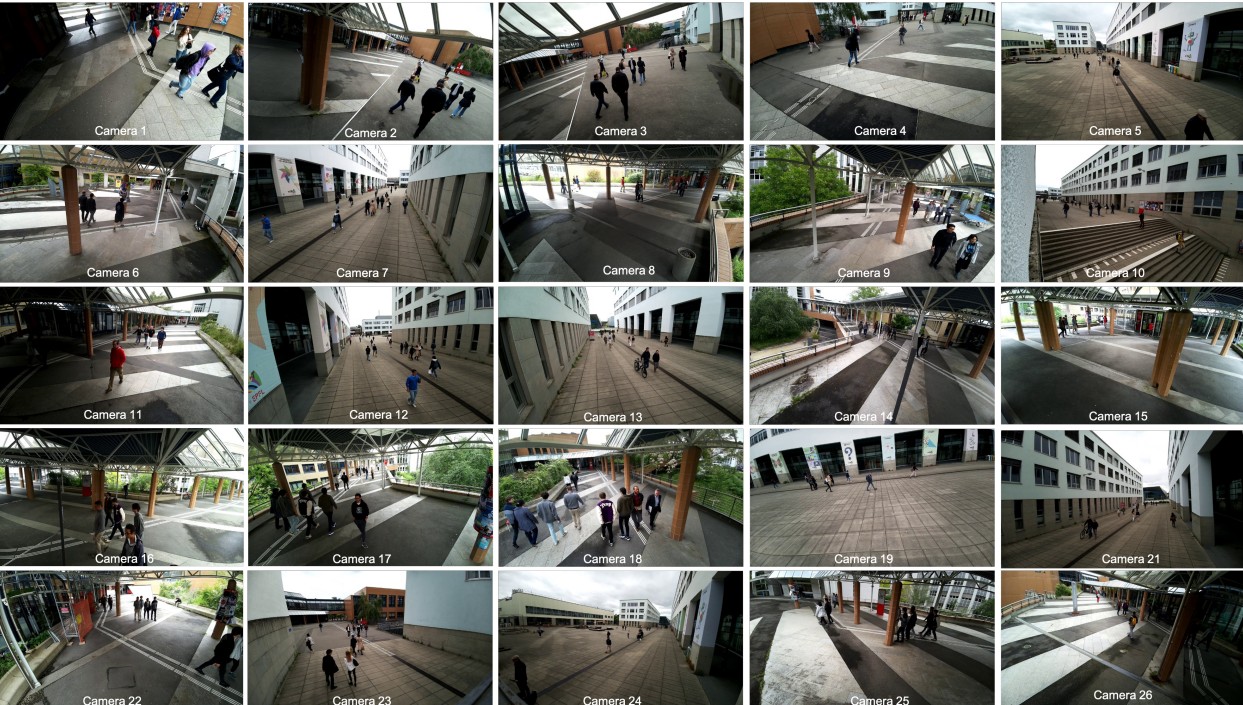

Figure 6: Example frames from our dataset showing views from all 25 cameras. The dataset contains 564k frames across these cameras, capturing 3k unique individuals over trajectories that span an average of 60 seconds and 80 meters.

SCOUT is also the only dataset that provides a structured 3D representation of the scene as a mesh, offering a richer spatial context for tracking and analysis. Fig. 6 presents example frames from the dataset.

### A.9 SCOUT Videos

The supplementary archive includes video visualizations of results on the single-view tracking task. We provide results for the baseline trackers ByteTrack and SORT, alongside the performance of our best model, UMPN+SP.

### A.10 Scene Occlusion Challenges

Our SCOUT dataset presents particularly challenging occlusion scenarios that distinguish it from existing tracking datasets. Fig. 7 illustrates two representative sequences that demonstrate the complexity of occlusions in our real-world multi-camera setup. The visualization shows how people are systematically occluded by architectural elements, creating both isolated and repetitive occlusion patterns that require sophisticated reasoning for robust tracking.

**Measuring Occlusion** Since the SCOUT dataset provides a 3D scene reconstruction, we can measure the occlusion of a detection by checking if the ray from the camera to the detection intersects the scene mesh before reaching the detection. We use the same method as in Section A.2.

**Occlusion Metrics** We measure the occlusion by computing the ratio of the number of occluded detections to the total number of detections among the final tracked detections. An increase in the occlusion rate indicates that the method is able to handle the occlusion better.

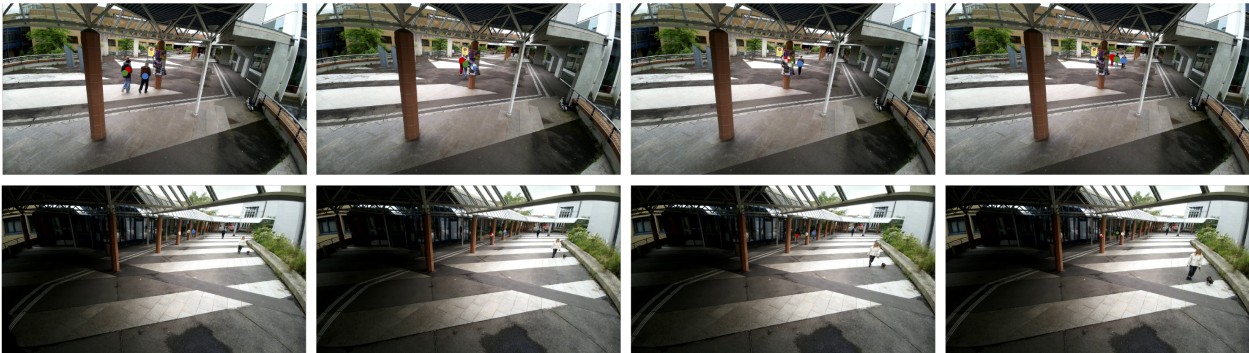

Figure 7: **Scene occlusion challenges in SCOUT dataset.** Two sequences of 4 frames each illustrate the challenging occlusion scenarios present in our dataset. People are marked with colored dots; when occluded, the dots display a checkerboard pattern. **Top sequence:** Basic occlusion scenarios where individuals become temporarily occluded by scene structures. **Bottom sequence:** More challenging repetitive scene occlusion as a group of people walks along columns, creating systematic and recurring occlusion patterns that are particularly difficult for tracking algorithms to handle.

| Method | Occ. Ratio (%)↑ | Num. Occ. Det. | Total Det. |
|---|---|---|---|
| GroundTruth | 5.20 | 88'235 | 1'697'809 |
| ByteTrack MV | 3.71 | 25306 | 682'129 |
| UMPN (Ours) | 4.0 | 19'342 | 480'786 |
| UMPN (Ours) + SP | **4.48** | 21'673 | 484'014 |

Table 10: **Occlusion handling evaluation on SCOUT dataset.** We measure the ability of different methods to successfully track occluded detections. The occlusion ratio represents the percentage of successfully tracked detections that were determined to be occluded by scene geometry. Higher ratios indicate better occlusion handling capability. SP stands for Scene Prior.

**Occlusion Results** We report the occlusion results in Table 10. When using scene priors we can see that the occlusion ratio increased from 4.0 to 4.48, indicating that using the scene prior is beneficial for handling scene occlusions. Both proposed models outperform the ByteTrackMV baseline.

