# OpenReview forum: "Uniﬁed People Tracking with Graph Neural Networks"
_TMLR — Rejected by TMLR_

### Review · Reviewer_nzSM · 2025-09-03

**Summary Of Contributions:**

1. Proposed a GNN-based MOT approach that operates on a spatio-temporal graph built from multi-view detections.
2. Introduced a new multi-view MOT dataset consisting of two 20-minute sequences captured from 25 different views.
3. Proposed novel types of vertices and edges for the graph, such as camera vertices and context edges, designed to improve occlusion reasoning.
4. Demonstrated competitive performance in both single-view and multi-view benchmarks within a unified framework.

There are many GNN-based MOT approaches in the literature, making some aspects of this work such as 1 and 4 feel incremental. Nevertheless, given TMLR’s acceptance criteria, I will not weigh novelty concerns heavily in my rating.

**Audience:**

Yes

**Audience Explanation:**

The paper introduces some novel elements, such as contextual edges and camera vertices for a GNN-based multi-view MOT approach, along with a new multi-view MOT dataset involving a large number of cameras. I believe these contributions are sufficient to interest the TMLR audience working on multi-view MOT, provided the findings are presented in a convincing manner.

**Broader Impact Concerns:**

The Broader Impact section is satisfactory in its current form.

**Claims And Evidence:**

Yes

**Claims Explanation:**

Claims checked against experimental results:
1. Benefit of multi-view compared to single-view through incorporation of scene structures
2. Benefit of dynamic learning using a dynamically-built spatio-temporal graph
3. Approach matches the state-of-the-art in single-camera setups and far exceeds it in multi-camera ones
4. Benefit of contextual edges and camera vertices to demonstrate improved occlusion reasoning

The authors demonstrated (1) and (2) in Tables 4 and 6, respectively.

Claim (3) is partially supported by Tables 2–4. However, it is unclear why the performance gap between the proposed method and ByteTrack is much larger on the new dataset (single-view setting, Table 4) than on the MOT Challenge benchmarks (Table 3). Clarification here would be helpful.

In addition, the baseline chosen for multi-view evaluation in the new dataset (a simple modification of ByteTrack) seems too weak. The authors note that prior works (Engilberge et al., 2023; Teepe et al., 2024) could not be run due to memory limitations. I suspect this is due to a large number of views offered by the new dataset? Could a comparison still be made using fewer views if that was the case? Outperforming the modified ByteTrack in the multi-view setting and framing this as “far exceeding the state-of-the-art” (page 2) does not feel well supported by the evidence.

Claim (4) is also not clearly supported. For example, in Table 4, the benefit of the scene prior via context edges for occlusion reasoning is minimal: the difference in IDF1 (a key indicator of occlusion handling) is less than 0.1 in the multi-view setting. Table 5 shows ablation results on contextual edges, but these are presented on MOT17, which is not a multi-view dataset, limiting the relevance of the evidence.

**Requested Changes:**

I recommend that the authors revise the manuscript to address my concerns regarding claims (3) and (4) that I listed above.

Other minor changes:
1. “Due to faults during the acquisition campaign, one of the sequences features only 22 cameras while all the others feature 25”: the phrase “all the others” here seems inconsistent with the earlier statement that the new dataset contains only two 20-minute sequences (page 1).
2. Typo — “taking their for vertex vi” (right above Eq. (4))


------------ Post-rebuttal -------------

Thank you for the rebuttal. I am satisfied with the authors’ responses and revisions addressing the concerns I raised. I also reviewed the major concerns from other reviewers and find the authors’ responses to be satisfactory. Therefore, I recommend acceptance of this draft for TMLR.

---

> ### Author Response · Authors · 2025-09-25
> **Response to Review**
>
> ## Claim 3
> The main reason for the performance gap between the proposed method and ByteTrack on the new dataset is that we use an off-the-shelf detector without any fine-tuning. In contrast, most MOT Challenge methods rely on detectors fine-tuned specifically on the MOT Challenge dataset. We did this because using an off-the-shelf detector without fine-tuning on SCOUT provides a more realistic generalization scenario. In this context, our intuition is that ByteTrack is more sensitive to detector performance, while our method is better able to cope with noisy detections.
>
> We agree that we cannot claim to exceed the state of the art on a new benchmark after comparing against only a few baselines and this was not our intention. Thus, our claim is not based on performance on SCOUT but rather on the performance on WILDTRACK on which many other methods have been tried. There,  the improvement is significant: An increase of 2.1 points in MOTA and 1 point in IDF1. Such improvements are meaningful as these metrics are already above 90 and 95, respectively.
>
> Nonetheless, we followed your suggestion and added new experiments comparing our method with MVFlow on a subset of the new dataset, using only 8 mostly overlapping views. The results are presented in Table 8. On this subset our methods outperform MVFlow, in both MOTA and IDF1.
>
> ## Claim 4
> In Table 4, the difference between UMPN and UMPN+SP does not lie in the use of contextual edges, (both models use them) but rather in the inclusion of camera vertices and edges. We have updated the caption to clarify this. To specifically assess the contribution of contextual edges in a multi-view setting, we have added a new experiment in a multi-view setting on Wildtrack, reported in the additional row of Table 6. Consistent with the results in Table 5, these findings further confirm the benefit of contextual edges.
>
> ## Other corrections
> - The typos and inconsistencies have been addressed; thank you for pointing them out.

---

### Review · Reviewer_iyeV · 2025-09-03

**Summary Of Contributions:**

1.	This paper proposes a unified approach that handles both single and multi-view tracking through dynamic spatiotemporal graphs.
2.	Graph with scene structure is proposed to integrate scenes prior.
3.	A new large-scale dataset comprising 25 camera views, featuring extensive occlusions and detailed scene reconstructions is developed.
4.	The proposed method achieves state-of-the-art performance on both single-view and multi-view people tracking benchmarks.

Strengths

- This paper constructed a new dataset for Multiview pedestrian tracking.
- This paper shows some ablation studies of the proposed method.

Weaknesses

- The writing is difficult to understand.
- Impact of occlusion (scene prior) is not evaluated.
- The proposed method may not be state-of-the-art.

**Audience:**

Yes

**Audience Explanation:**

Object tracking would be of limited interest but cannot deny all individuals’ interests.

**Claims And Evidence:**

No

**Claims Explanation:**

- ``State-of-the-art performance without sequence tuning. “ . It is not clear what is referring the “sequence-specific tuning.”
I found SUSHI [11] shows better performance than the proposed method on MOT17 and MOT20.

- Examples of occlusion are not shown.
- Occlusion robustness is not evaluated.

**Requested Changes:**

Show the examples of occlusion.
Visualization of tracking results with/without occlusion handling.
Evaluate the impact of scene prior and occlusion handling.

Compare with more state-of-the-art methods.

Rewrite the paper more understandable and self-contained.

-	The title is too broad. This paper is not the first work that use Graph Neural Network for object tracking. It is not clear if existing methods cannot handle both single and multiview tracking.
-	P1. Abstract  "without relying on pre-computed trackless”. Do existing tracking-by detection methods rely on the pre-computed trackless?
-  ``Existing tracking benchmark focus on short sequences with minimal scene occlusion.”.
However, I can find papers of occlusion handling.
https://scholar.google.co.jp/scholar?hl=ja&as_sdt=0%2C5&q=multi+camera+tracking+occlusion+handling&btnG=
-	Three MLPs in Eq.(1) seems to be different since the dimensions of $p_d$, $w_d$, $t_d$.

- P.16 $b_{src}$ indicates if the detection originated from the camera. Why can this information be determined without tracking results?

- Training and test splits of SCOUT is not mentioned, and it is not clear why the ground-truth edge/vertex can be used.

-	Improve reference. For example, P14. [20] and [21] are the same. Some references are incomplete.

---

> ### Author Response · Authors · 2025-09-25
> **Response to Review**
>
> ## On State-of-the-art
> As explained in Section 2, paragraph *Learning-based Methods*, SUSHI [11] is an offline tracking method, meaning it can only process the entire video at once. A key contribution of our work is a dynamic learning strategy (Section 3.3) that allows a GNN-based method to be used online, as opposed to offline. Since SUSHI relies on future frame information to associate detections in previous frames, whereas our approach does not, its improved performance is not truly relevant. Additionally, SUSHI is a single-view tracking method, while our approach generalizes to both single-view and multi-view tracking.
>
> ## Example of Occlusion:
> We have added examples of scene occlusions present in the new dataset to the supplementary material (see Figure 7).
>
> ## Evaluation of Occlusion Robustness:
> We emphasize that our contribution is not specifically aimed at occlusion handling, but rather at demonstrating the broader benefits of incorporating scene priors for tracking. Occlusion is mentioned only as an intuitive example of why such priors can be helpful. Our dataset naturally contains challenging occlusions, which enables us to investigate this aspect. While Table 4 already reports an ablation of the scene prior, in response to the reviewer’s comment we additionally provide a dedicated evaluation of scene-induced occlusions in Section A.10.
>
> # On Requested Changes:
> ## Title
> We are clearly not the first ones to use Graph Neural Networks for people tracking. However, we believe we are the first to do it in an online manner that generalizes to both single-view and multi-view scenarios. Our title was meant to reflect that. We could change it to something like “One Graph to Track Them All: Dynamic GNNs for Single- and Multi-View Tracking” to be more specific.
>
> ## Precomputed tracklets
> Tracklets have been, and still are, extensively used to people tracking and re-identification. Below are three references, but we agree that modern methods are using them less and less and have adjusted the introduction accordingly.
>
> Andriluka, Mykhaylo, Stefan Roth, and Bernt Schiele. "People-tracking-by-detection and people-detection-by-tracking." 2008 IEEE Conference on computer vision and pattern recognition. IEEE, 2008.
>
> Li, Minxian, Xiatian Zhu, and Shaogang Gong. "Unsupervised tracklet person re-identification." IEEE transactions on pattern analysis and machine intelligence 42.7 (2019): 1770-1782.
>
> Kim, Jeongho, et al. "Cluster self-refinement for enhanced online multi-camera people tracking." Proceedings of the IEEE/CVF Conference on Computer Vision and Pattern Recognition. 2024.
>
> ## On occlusions
> The reviewer is right to point out that occlusions are a well-known challenge in tracking, with numerous methods developed to address it. However, when examining both existing approaches and publicly available datasets, one can see that the focus is primarily on people occluding each other, which typically occurs when their density increases. In contrast, occlusions caused by objects and structures present in the environment, such as buildings and trees, have received comparatively little attention. SCOUT aims to remedy that. The selected location, depicted in the new Figure 7, was chosen with this goal in mind.  The availability of multiple views and complete scene geometry should enable a more in-depth exploration of occlusion caused by the environment than has been possible so far.
>
> ## Clarifications
> - The reviewer is  right.  There are indeed three distinct MLPs in Eq. (1). We initially omitted the exact parametrization of each MLP for simplicity, but we have now added the full details.
>
> - **b_src:** As explained in the first paragraph of Section 3.1, detections are obtained by running an off-the-shelf detector on each frame. Therefore, the originating camera of each detection is already known, without the need for tracking results.
>
> - The ground truth edges/vertices are used only during training for supervision or during evaluation to compute metrics. They are not used or required during inference. We have updated the manuscript to include the train/test splits used to train and validate SCOUT in Section 5.3.
>
> - We have corrected the erroneous references. Thank you for pointing this out.

---

### Review · Reviewer_5SZQ · 2025-09-16

**Summary Of Contributions:**

This paper proposes a GNN-based framework to tackle association within Tracking-by-detection-based Multi-Object Tracking (MOT) in both single and multi-camera setups. The main methodological contribution is a graph structure consisting of several types of edges, including edges across time, views and space, in addition to 'camera edges', capable of handling multi-view scenarios. The proposed graph is coupled with an efficient heuristic algorithm for online inference. In addition, the paper introduces a new large-scale multi-view MOT dataset, with significantly more cameras and longer duration than previous works.

Main strengths:
- Introduces a novel graph-based  framework to tackle both single and multi-camera association.
- This method yields strong results compared to previous online methods across benchmarks, and appears to be computationally efficient at inference, making it practical.
- Introduces a potentially valuable dataset for multi-camera tracking.

Main weaknesses:
- Serious presentation problems. E.g. the introduction contains multiple questionable claims, about the state SOTA methods (being heuristic) and existing datasets (being focused on short sequences)
- The introduced framework borrows methodology heavily from previous graph-based work (e.g. Braso et. al [9]). However, no direct comparison between this work and MPNTrack and other follow-up graph-based work (e.g. Cetintas et al. [11]) is provided. More over, significant parts of the methodology text (e.g. Section 3.2) do not credit previous work [9] despite heavily borrowling its methodology.
- Questionable value of dataset introduced: it seems to contain very low density of pedestrians compared to other works, raising doubts about its tracking difficulty/complexity. (e.g. 10x the IDs of WILDTRACK despite having 200x more frames (or 40 if normalizing by FPS)). So overall 1/4 of the pedestrian density. The difference is even more extreme if comparing to the MOT17 or MOT20 datasets.
- The proposed method cannot tackle long occlusions despite focusing its introduced dataset on clip length.

**Additional Comments:**

- Why are edges initialized with zeros, as opposed to e.g. the features used in previous work (e.g. MPNTrack, SUSHI)? Was this ablated?
- Does the provided dataset annotate only cylinders or oriented 3D bounding boxes?
- I'm confused about the explanation in Section 3.3. When applying the sliding window, are node/edge embeddings from the previous (overlapping) window used for initialization in the next 'window'? If so, this needs to be made more clear in section 3.2. Otherwise, claims such as 'the model learns to smoothly accumulate information over long sequences' do not appear to make sense.

**Audience:**

Yes

**Audience Explanation:**

Despite its presentation and experimental weaknesses, this work tackles Multi-Object Tracking in a promising direction: with a unified graph formulation capable of tackling both single and multi-camera tracking. The method shows empirically good performance and is fast as inference, hence having potential to be a very practical method. Moreover, the unified graph formulation is generally principled and elegant. Overall, this method has the potential to be a valuable addition for the MOT task, a fundamental problem in computer vision.

**Broader Impact Concerns:**

No concerns.

**Claims And Evidence:**

No

**Claims Explanation:**

While some of the claims are indeed supported (e.g. good performance in benchmarks), some others are not:
- Introduction paragraph 1: SOTA being dominated by heuristic methods. Multiple counter-examples exist, including transformer-based (e.g. MOTR [Zeng et al.], Trackformer [Meinhardt et al.) and GNNs (e.g. SUSHI [Cetintas et al.].
- Introduction paragraph 2: 'existing tracking benchmarks focus on short sequences with minimal scene occlusions'. This claim is essentially false: benchmarks such as MOT20 contain sequences with over a minute length and with extreme highly dense occlusions, in fact significantly more present than in the proposed benchmark.
- Value of scene priors: the proposed camera edges (contribution 2) do not appear to be ablated in experiments. Moreover, I question their need: why are nodes and edges required, instead of simply adding them as node attributes?
- Value of proposed dataset (contribution 3): the authors claim presence of 'extensive occlusions' but no analysis on them is provided.
- The performance of the proposed 'unified model' is hard to assess since no direct comparison is provided w.r.t previous graph-based 'non-unified' methods.

**Requested Changes:**

Critical adjustments:
- Fixing introduction to accurately reflect state of current tracking literature, both in terms of algorithms and datasets.
- Either toning down claims about proposed dataset or proposing an analysis showing that it actually contains long occlusions, and justifying the low density numbers in Table 1.
- An ablation on the role of camera node/edges and whether node features for them would suffice.
- Providing credit in e.g. Section 3.2 to previous work e.g., [9] upon which this work is heavily based on.
- Providing a direct comparison to MPNTrack and follow-up graph-based work. In principle, this work can also be evaluated offline by simply adapting the heuristic postprocessing algorithm, making this work directly comparable. If this is not possible, a strong justification is required. Overall, this work would benefit significantly to showing further versatility to offline setups.

Other, less critical improvements:
- An analysis on the characteristics (occlusion length, density, etc.) of the provided dataset
- Improving Table 1 by including e.g. number of clips, actual clip length in seconds (not only fps and frames).
- Improving the explanation in Section 3.3 (see my Additional Comments)
- Why isn't larger temporal context helpful? Any intuition?

---

> ### Author Response · Authors · 2025-09-25
> **Response to Review**
>
> # On Requested Changes:
>
> ## Introduction:
> We have revised the introduction to better reflect the current state of tracking literature and to clarify that our claims about occlusions refer to **scene-structure-induced occlusions**, not those caused by other people. We agree that existing benchmarks extensively feature the latter but note that the former are not represented.
>
> Regarding  the distinction between a short and a long sequence,  that is somewhat subjective. Nevertheless having sequences that are, on average, nearly 18 times longer than those in MOT20 is a significant difference worth highlighting. We have updated Table 1 with the requested information.
>
> ## Camera Node/Edge
> Camera edges are introduced in Table 4, under the label +SP (scene prior).  When embedding the scene prior into the graph, we initially considered using node features. However, we eventually settled on adding explicit camera nodes and edges because it offered greater flexibility in terms of memory usage and avoided duplicating the same information multiple times.  Since a single camera is shared across all detections visible from it, it was then natural to also share the corresponding features during the update process.
>
>
> ## Prior work
> We have revised the introduction and Section 3.2 to provide clearer acknowledgment of prior work.
>
> ## Comparison to offline methods
> We appreciate the reviewer’s suggestion. However, adapting our algorithm to an offline setup would go against the main motivation and design principles of our approach, which is specifically optimized for online operation. While in principle one could redesign the post-processing to mimic an offline version, this would not constitute a fair comparison: offline methods have access to the full graph, whereas our dynamic graph is inherently limited to a subset of the sequence.
>
> For this reason, we believe that comparing offline and online methods directly is not particularly meaningful, as they target different scenarios. Nonetheless, to give additional perspective, we could update Table 3 to include representative offline graph-based methods (e.g., MPNTrack, SUSHI). This makes it explicit that our online method already achieves performance close to state-of-the-art offline approaches. For example, on MOT20 our method reaches 77.7 MOTA compared to 74.3 for SUSHI, and 75.2 IDF1 compared to 79.8 for SUSHI. The IDF1 gap is expected, as preserving identity is naturally easier when the entire sequence is available, but the results highlight that our approach remains highly competitive despite operating under stricter online constraints.
>
> ## Occlusions
> We have added Figure 7 to illustrate the types of scene occlusions present in SCOUT.
> In addition, we included an experiment in Section A.10 and Table 10 to evaluate the effect of scene occlusions on our methods.
>
> ## Clarification on Section 3.3
> There seems to be a misunderstanding of our contribution.  As stated in Section 3.3 (Dynamic Learning), our method relies on a **single dynamic graph**.  The term *sliding window* refers only to the detections currently considered, unlike in previous methods that divided a sequence into multiple static graphs.
>
> In our approach, a single graph is maintained; new detections are added while older ones are removed. Consequently, there is no need to reinitialize nodes already present in the graph, initialization is required only for newly added nodes. This principle applies consistently during both training and inference.
>
> # Additional Comments
> ## Temporal Context
> We found that a larger temporal context is beneficial, although during training we are limited by GPU memory.
>
> The Table 5 results on the maximum temporal distance were obtained by modifying this value at inference time (the model itself was trained with a value of 4). These results show that using a longer temporal distance (6) than during training can still improve performance.
>
> We believe the negative results observed when increasing it further (8) are due to a mismatch between training and validation conditions. If computationally feasible, training the model directly with a maximum distance of 8 would likely lead to additional improvements. We have clarified this in the manuscript.
>
> ## Edge initialization
> We tried initializing edges instead of nodes and found it degraded performance, compared to node initialization.  We believe this is due to the nature of our training, where gradients are accumulated over multiple sliding windows before being updated. By initializing edges to zero, the model is allowed to learn and determine the most suitable representation for the edge features.
>
> ## Dataset annotation
> The annotation tool represents 3D annotations as cuboids; however, the orientation was not adjusted, making them more similar to cylinders.  Since trajectories are also provided, it should be possible to derive reasonable orientations for the cuboids.

---

### Author Response · Authors · 2025-09-25
**General comment**

We would like to sincerely thank the reviewers for their valuable feedback and insightful comments. We have carefully considered each point and have revised the paper accordingly, including the suggested improvements and additional analyses. The revisions are highlighted in red for your convenience. We appreciate the time and effort invested in reviewing our work and look forward to addressing any further concerns.

---

### Decision · Action_Editor_mvch · 2025-11-30

**Recommendation:** Reject

**Audience:**

Yes

**Audience Explanation:**

This paper utilized and extended the methodology in [9] to online tracking by using dynamic graphs, which is interesting to folks working on multi-target tracking and other online uses of graphs.

**Claims And Evidence:**

No

**Claims Explanation:**

Reviewer 5SZQ made a good case for rejection: the dataset is simple and only considered occlusions by the scene context, rather than by other pedestrians. This is considered easy for the MOT task and underwhelms the current state-of-the-art of multi-target tracking datasets.  Besides, it is short to consider a 4-8 frame window for occlusion reasoning by the standards of 2025. Finally, the baselines used in the paper are weak and the amount of baselines is small, and it didn't compare with other online multi-target tracking methods. The other two reviewers have leaning accept recommendations but the arguments are very weak and the tones are leaning toward being not very interested in this paper. Hence, the associate editor believes that the quality of the work falls short of TMLR standards.

---

> ### Author Response · Authors · 2025-12-04
> **No comment visible**
>
> Dear AE,
>
> I am a little confused. No comments, positive or negative, are visible on my console. The only thing I can see is the remark of Reviewer nzSM who declares himself "satisfied with the authors’ responses and revisions addressing the concerns I raised". Could we see the other comments, and perhaps, be given a chance to respond?
>
> Cheers
> Pascal